# Stable and solubilized active Au atom clusters for selective epoxidation of *cis*-cyclooctene with molecular oxygen

Linping Qian[1,2,*], Zhen Wang[2,3,*], Evgeny V. Beletskiy[2], Jingyue Liu[4], Haroldo J. dos Santos[2,5,6], Tiehu Li[3], Maria do C. Rangel[5], Mayfair C. Kung[2] & Harold H. Kung[2]

The ability of Au catalysts to effect the challenging task of utilizing molecular oxygen for the selective epoxidation of cyclooctene is fascinating. Although supported nanometre-size Au particles are poorly active, here we show that solubilized atomic Au clusters, present in $ng\,ml^{-1}$ concentrations and stabilized by ligands derived from the oxidized hydrocarbon products, are active. They can be formed from various Au sources. They generate initiators and propagators to trigger the onset of the auto-oxidation reaction with an apparent turnover frequency of $440\,s^{-1}$, and continue to generate additional initiators throughout the auto-oxidation cycle without direct participation in the cycle. Spectroscopic characterization suggests that 7–8 atom clusters are effective catalytically. Extension of work based on these understandings leads to the demonstration that these Au clusters are also effective in selective oxidation of cyclohexene, and that solubilized Pt clusters are also capable of generating initiators for cyclooctene epoxidation.

[1] Department of Chemistry and Shanghai Key Laboratory of Molecular Catalysis and Innovative Materials, Fudan University, Shanghai 200433, China. [2] Chemical and Biological Engineering Department, Northwestern University, Evanston, Illinois 60208, USA. [3] School of Materials Science and Engineering, Northwestern Polytechnical University, Xi'an, Shaanxi 710072, China. [4] Physics Department, Arizona State University, Tempe, Arizona 85281, USA. [5] Instituto de Química, Universidade Federal da Bahia, Campus Ondina, Federação, Salvador 40170-280, Bahia, Brazil. [6] Instituto Federal de Educação, Ciência e Tecnológica da Bahia, Eunápolis 45820-970, Bahia, Brazil. * These authors contributed equally to this work. Correspondence and requests for materials should be addressed to M.C.K. (email: m-kung@northwestern.edu) or to H.H.K. (email: hkung@northwestern.edu).

Current interest in noble metal atom clusters with dimensions less than the Fermi wavelength of conduction electrons arises from their distinctly different physical and chemical properties from those of single-atom species and bulk metal, and their potential to provide linkages between the materials at both ends of the size spectrum. They exhibit electronic quantum size and photochromic effects[1], a high degree of structural fluxionality[2–4], and cluster size-sensitive catalytic properties, for example, ref. 5. In addition to possessing fascinating properties, they also have broad applications in diverse fields such as sensors and bioimaging[6]. Consequently, significant effort has been expanded in the preparation and characterization of well-defined materials such as size-selected clusters[7] or ligand-capped monodisperse nanoclusters[8]. Colloidal preparation is also common, and the strongly bound surfactants are critical in determining the morphology and size of the colloidal particles[9]. Although effective in atom cluster preparation, strong metal–ligand interaction could influence the surface chemical properties of the clusters, especially their catalytic properties. Thus, it remains important to investigate properties of these atom clusters under conditions relevant to their applications, such as under catalytic reaction conditions.

Epoxides are precursors to many valuable chemicals, and their direct selective formation from olefins using $O_2$ in a solvent-free reaction is environmentally friendly, highly desirable but challenging. Selective epoxidation of olefins using $O_2$ is practiced only for ethylene and higher olefins that do not contain allylic hydrogen atoms[10–12]. For other olefins, expensive oxidants such as $H_2O_2$ or alkyperoxides are used. Thus, the seminal work of Lambert demonstrating the feasibility of converting styrene to its epoxide (10–30% selectivity) using $O_2$ over supported Au catalysts derived from $Au_{55}$ clusters generated considerable excitement[13]. Subsequently, the use of $O_2$ to effect epoxidation of cyclic olefins and styrene has been explored for a diverse set of Au catalysts deposited on a variety of supports[8,14–16], but the selectivities, except with cis-cyclooctene (COE) substrate, were poor. Even for COE, the oxidation activity was low[17].

Improvement of the Au catalysts in these reactions necessitates an understanding of what constitutes the active catalyst. The examples mentioned above were all supported Au catalysts, yet there appears to be no consensus regarding the optimal particle size of Au[8,13–15,17]. The Au particles in these examples ranged from monodisperse atomic clusters[8] to polydispersed Au nanoparticles spanning a broad size range of 20–150 nm (ref. 14). These catalysts were considered to be heterogeneous in nature and their apparent recyclability[11,15] reinforced this impression. However, the reused catalysts were recovered by centrifugation[11,15], a procedure whereby solubilized Au species can be re-deposited onto the support. Thus, the seeming lack of consensus on the optimal Au particle size[8,13–15,17] can be reconciled if we consider the catalysts as only a source of Au and the catalytic relevant species are solubilized Au atom clusters. Here we demonstrate the presence of solubilized, stable Au atom clusters containing less than 10 atoms that are generated in the reaction process and stabilized by the reaction medium. We demonstrate that they are active in initiating cyclooctene epoxidation with an apparent turnover frequency of $440 \, s^{-1}$. We further correlate the appearance of these atomic clusters with distinctive optical properties to the emergence of high catalytic activity and show that solubilized Pt atom clusters are also effective in initiating this reaction.

## Results

**Generation of solubilized Au clusters.** Solvent-free oxidation of COE using $O_2$ was conducted in the absence of light and either with or without COE stabilizer present. Removal of the COE stabilizer was achieved either by distillation or KOH treatment (see Supplementary Methods Section 1.1 for details), having in mind the recent report that oxidation of commercial cyclic alkenes using $O_2$ over Au/graphite was only possible after removal of the added stabilizer[17]. The former method seemed to yield more consistent results. Using the Au/SiO$_2$-A catalyst, the stabilizer-free COE was oxidized to cyclooctane epoxide with high selectivity ($\geq 80\%$) after an induction period of $\sim 2 \, h$ (Fig. 1). The induction period was substantially longer ($>5 \, h$) for two other Au/SiO$_2$ preparations (Au/SiO$_2$-B1 and -B2) of average Au particle sizes of 0.9 and 4.5 nm, in which the Au particles were encapsulated by a nanoporous layer of SiO$_2$ (Supplementary Fig. 1). Starting with AuCl or AuCl$_3$, similar conversion-time profiles but with significantly longer induction periods were observed (Fig. 1). In these data shown for AuCl, the stabilizer was not removed as its presence did not affect the induction period. The Au salts are poorly soluble in COE and the gas disperser and reaction vessel walls were partly coated with a Au film at the end of the reaction, likely formed from the decomposition and reduction of the unstable $(cis\text{-}C_8H_{14})_x AuCl_y$ complex[18].

Despite the large differences in the induction periods, the product distributions using any of the Au sources were very similar (Supplementary Table 1), and the post-induction reaction rates were also similar to within a factor of 2 (Supplementary Table 2). In spite of the uncertainties in correction for evaporative losses, the carbon balance was good, better than $\pm 10\%$ consistently even at $>70\%$ conversion and as low as $\pm 1\%$ at $<10\%$ conversion. The predominant oxidation product was cyclooctane epoxide (80% selectivity at 50% conversion, Supplementary Fig. 1 and Supplementary Table 1), followed by small amounts of 2-cyclooocten-1-ol, 2-cyclooocten-1-one and trace amounts of 1,2-cyclooctanediol. Small amounts of cyclooctene 3-hydroperoxide were detected by nuclear magnetic resonance (NMR)[19]. A small amount of high molecular weight products, whose sticky nature prevented the use of electrospray ionization mass spectrometry to characterize the reaction mixture, was also present. As a control experiment, without adding any Au source, there was no reaction detected for stabilizer-free COE for over 10 h, and slow reaction afterwards forming epoxide (curve d, Fig. 1). Thus, Au is effective in initiating the COE epoxidation reaction.

There is literature precedence of an induction period in cyclic alkene oxidation with $O_2$. This was ascribed to the time needed

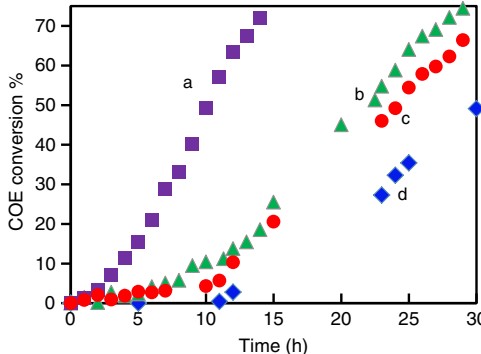

**Figure 1 | Time course of cyclooctene conversion.** COE conversion using 80 mg Au/SiO$_2$-A (a, square), 6 mg AuCl (b, triangle), 7 mg AuCl$_3$ (c, circle), or no Au (d, diamond). Reaction conditions: 10 ml COE, 1 ml decane, $O_2$ bubbling rate 30 ml min$^{-1}$, 100 °C. For experiments with Au/SiO$_2$ or when no Au was present, COE was stabilizer free but, for experiments with AuCl and AuCl$_3$, COE contained 100–200 p.p.m. Irganox 1,076 stabilizer. Conversion is defined based on initial COE present.

for the supported Au particles to attain a sufficiently large size ($>2$ nm) for activity[15]. However, we observed that the reaction rates after the induction period were similar regardless of the duration of the induction period. At 50% conversion, the rates were 5.0, 3.9 and $3.5 \pm 0.2$ mmol h$^{-1}$ for Au/SiO$_2$-A, Au/SiO$_2$-B2, and AuCl, respectively (Supplementary Table 2). These results suggested that the nature of the solid Au material is unimportant. In fact, the reaction proceeded equally well after the solid Au material was removed by filtration, *vide infra*. Thus, we propose that in the earliest phase of the induction period, a very small amount of hydrocarbon was oxidized and the oxygenated products served as ligands that extracted and solubilized active Au atom clusters into solution. These active Au atom clusters activated O$_2$ to abstract H atoms from COE molecules and initiated the auto-oxidation phase of COE conversion to epoxide. Indeed, analysis of the reaction solution after hot filtration using a 0.2 µm syringe filter to remove all solids showed low concentrations of solubilized Au typically from 10 to 200 ng ml$^{-1}$, depending on the Au source and reaction condition. Although the filtration was conducted as rapidly as possible, some cooling of the hot liquid and possible precipitation in the syringe was unavoidable, although none was observed. The formation of stable Au clusters requires both solubilization of Au and generation of stabilizing ligands. Both processes require O$_2$ and no reaction was detected when O$_2$ was replaced by N$_2$ in the gas stream. Similarly, formation of stable Au clusters can only be achieved in the presence of appropriate ligating molecules. Thus, the induction period was unchanged when O$_2$ was bubbled though a suspension of Au precatalyst in decane before mixing in COE.

That both oxygen and COE were necessary to generate solubilized Au suggests that the effective ligands that facilitate Au dissolution are oxygenated hydrocarbons. Addition of cyclooctanol, a molecule similar to the minor product cyclooctenol, to the reaction mixture did not affect the induction period or the reaction rate (Supplementary Fig. 3a). On the other hand, addition of cyclooctane-1,2-diol both shortened the induction period and enhanced the reaction rate, the extents of which increased with increasing concentration of diol added. Concurrently, the concentration of solubilized Au increased substantially (Supplementary Fig. 3b and Supplementary Table 3). Since the diol is oxidized readily to carboxylic acid under the reaction conditions, as determined by NMR, it appears that diacids were effective chelating ligands in stabilizing Au clusters.

**Catalytic role of solubilized Au clusters**. Aliquots of reaction-derived solution obtained by hot filtration were active for initiation of epoxidation of fresh COE, and the activity was stable for weeks when stored in the dark at room temperature. The reaction profile using a filtrate was similar to that of the original mixture but with an important distinction that the induction period was absent (Supplementary Fig. 4). Whether the reaction was initiated with an aliquot of filtrate or solid Au precatalysts, the reaction rate beyond $\sim 20\%$ conversion was similar (Supplementary Fig. 4) and significantly higher than those at low conversions. This phenomenon is consistent with the auto-oxidation nature of this phase of reaction.

After initiation, auto-oxidation of hydrocarbon is propagated by reactive intermediates such as hydroperoxide and other free radicals. In all reactions terminated at conversions $>60\%$, hydroperoxide was detected by titration at concentrations equivalent to $\sim 2\%$ of the initial COE concentration ($\sim 2\%$ yield). In one experiment using AuCl as the Au source, the concentration of hydroperoxide was monitored throughout the reaction and found to be equivalent to a yield of 0.9 at 28% conversion, 1.5 at 47% conversion and $\sim 2.0\%$ at conversions 60% and higher.

Together with solubilized Au, hydroperoxide and reactive intermediates in the reaction-derived filtrate contributed to the initiation of COE epoxidation and the elimination of the induction period. To differentiate and quantify the contribution of solubilized Au clusters to the initial reaction rate, the following experiments were conducted. Since it is known that triphenylphosphine (PPh$_3$) reacts quantitatively and converts peroxide to the corresponding alcohol[20], we first tested whether PPh$_3$ was able to quantitatively suppress the epoxidation reaction. Thus, a COE epoxidation reaction was initiated using *tert*-butyl peroxide (*t*-BuOOH) without any Au addition. After reaching 77% conversion (Supplementary Fig. 5), the solution was hot-filtered for procedural consistency although there were no solids. This stock filtrate obtained contained 2.95 mmoles of hydroperoxide, which was probably cyclooctene 3-hydroperoxide as the original *t*-BuOOH should have been consumed completely. A 2 ml aliquot of this stock filtrate was used to initiate another COE reaction (Fig. 2, curve c), and the initial rate was $0.75 \pm 0.25$ mmoles h$^{-1}$. To another 2 ml of the same stock filtrate, 1.05 equivalent PPh$_3$ was added, and the mixture was used to initiate another COE reaction. No peroxide was detected after the addition of PPh$_3$ and concomitantly, the initial activity was completely suppressed (Fig. 2, curve d). Thus PPh$_3$ was effective in consuming free radical initiators of the auto-oxidation reaction. These serve as control experiments for Au-catalysed initiation experiments.

A stock solution of a Au/SiO$_2$-A derived filtrate was first prepared. Then 2 ml of this stock solution was used to initiate a COE epoxidation reaction, and the initial rate was found to be $2.7 \pm 0.25$ mmol h$^{-1}$ (Fig. 2, curve a). Next, another 2 ml of the same stock solution was mixed with 1.05 eq. PPh$_3$ (with respect to hydroperoxide detected). This mixture, with no detectable hydroperoxide, was used to initiate another COE reaction. The initial rate was $1.8 \pm 0.25$ mmol h$^{-1}$ (Fig. 2, curve b), and the time course of product distribution was similar to that without PPh$_3$ addition (Supplementary Fig. 6). Thus, Au contributed 1.8 mmol h$^{-1}$ to the initial COE epoxidation rate, corresponding to a turnover frequency (TOF) of 440 s$^{-1}$ based on total

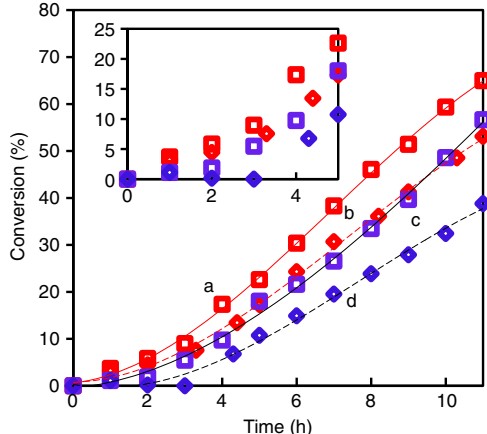

**Figure 2 | Effect of triphenylphosphine on reaction rates.** Effect of triphenylphosphine (PPh$_3$) addition on the initial rates of COE oxidation. Curves a,b: fivefold diluted filtrates obtained from COE oxidation using Au-SiO$_2$-A; Au = 20 ng ml$^{-1}$ and hydroperoxide = 0.29 mmoles. Curve a: filtrate, b: filtrate + 0.30 mmoles PPh$_3$. Curves c,d: fivefold diluted filtrates obtained from COE oxidation using *t*-BuOOH as reaction initiator. c: filtrate and d: filtrate + 0.62 mmoles PPh$_3$. Inset uses the same symbols.

solubilized Au present. The difference between the two rates ($0.9\,mmol\,h^{-1}$) was ascribed to contributions from hydroperoxide and other reactive intermediates.

The conclusions from these experiments were substantiated using other sets of stock filtrates. As shown by Exp. 1 in Supplementary Table 4, after removal of hydroperoxide with $PPh_3$, the stock Au filtrate retained significant activity to initiate the reaction, whereas the stock $t$-BuOOH filtrate lost its activity (Exp. 3).

An additional experiment was conducted to probe the effect of solubilized Au concentration on the initiation rate (Exp. 2 in Supplementary Table 4). Starting with a stock Au filtrate, a series of three consecutive runs was performed, in which 2 ml aliquot of the filtrate of the preceding run was used as initiator. Thus, at the start of each subsequent run in the series, the organic reactive intermediate would be relatively similar, but the solubilized Au would be repeatedly diluted, such that its concentration in Exp. 1 was 5.4 times more concentrated than the next run, and 29 times more concentrated than Exp. 2 (which was 157 times more dilute than the stock filtrate). In addition, 1 eq. of $PPh_3$ corresponding to the hydroperoxide present in the aliquot was added to Exps 1 and 2 at the beginning of the runs. The initial rate observed for Exp. 2 decreased by $\sim 25$ times compared with Exp. 1 (Supplementary Table 4). This very low initial rate in run with highly diluted solubilized Au is in line with experiments using stock $t$-BuOOH filtrate with $PPh_3$ addition, where there were no Au and the initial activity was not measurable.

Another series of experiments was performed to probe the effect of solubilized Au concentration on the rate of auto-oxidation. In these experiments (Fig. 3), a standard COE reaction using Au/SiO$_2$-A as the Au source was conducted first (Run 1). Then 2 ml of its filtrate was used to initiate Run 2. Afterwards, 2 ml of the solution from Run 2 was used to initiate Run 3. As before, the concentrations of organic hydroperoxide and reactive intermediates were similar at the beginning of Runs 2 and 3, but the concentration of solubilized Au in Run 3 was 5.4 times lower than Run 2, and 29 times lower than Run 1. No induction period was observed in either Run 2 or 3 as observed earlier, and the reaction rates at 50% conversion decreased from $5.3 \pm 0.2\,mmol\,h^{-1}$ in Run 1, to $4.6 \pm 0.2\,mmol\,h^{-1}$ in Run 2, and $3.5 \pm 0.2\,mmol\,h^{-1}$ in Run 3 (Supplementary Table 5). This small but consistent decrease in rate with increasing dilution of

Au was observed reproducibly, and is consistent with an auto-oxidation mechanism that does not depend on Au, and that the function of Au is to initiate the reaction. The slight decrease in reaction rate with each subsequent dilution of Au indicated that, although not a catalyst in the auto-oxidation cycle, the solubilized Au continued to contribute to the COE conversion by generating initiators for the auto-oxidation cycle. As the concentration of Au decreased with dilution, this contribution diminished, and the auto-oxidation reaction rate declined. For the experiment under consideration, there was 6 nmol solubilized Au in the original solution. If we assume that after 29-fold dilution, the contribution of solubilized Au to the auto-oxidation reaction became negligible, then the solubilized Au was contributing $0.33 \pm 0.2\,mmol\,(nmol\,Au\cdot h)^{-1}$ to the auto-oxidation reaction by injecting initiators at 50% conversion, or an apparent TOF of $92\,s^{-1}$.

**Nature of the solubilized Au clusters**. The filtrate from a reaction using the Au/SiO$_2$-A as the Au source was examined with aberration-corrected transmission electron microscopy (TEM; Fig. 4). The images showed Au species present mainly as small clusters up to $\sim 0.7$ nm, including single Au atoms. A few larger clusters were found occasionally. Supplementary Fig. 7 shows the size distribution of the Au clusters.

These filtrates were also examined using fluorescence spectroscopy. The filtrate from reaction using the Au/SiO$_2$-A source showed a broad emission peak at $\sim 450$ nm with a corresponding excitation peak at $\sim 405$ nm (Fig. 5, curves a, a′), which were not observed without Au (that is, not present in the reaction mixture derived from $t$-BuOOH initiator, Supplementary Fig. 8). Similar spectra were observed with filtrates derived from AuCl (Supplementary Fig. 9) and AuCl-diol combination. For the latter, the concentration of solubilized Au increased with diol/Au ratio, and the intensities of both emission and excitation spectra increased simultaneously (Supplementary Fig. 10).

Small Au atomic clusters are known to exhibit discrete absorption and fluorescence, and the emission maxima have been shown to scale with the cluster size as predicted by the spherical Jellium model[21]. On the basis of this model, these spectra indicated the presence of predominantly eight-atom clusters (Supplementary Table 6). These emission peaks were not

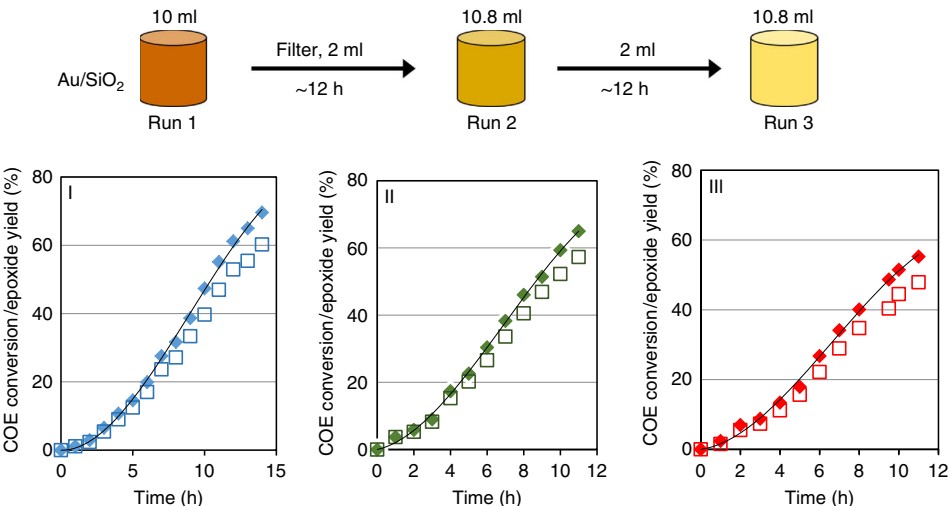

**Figure 3 | Cyclooctene conversions in consecutive experiments.** Experiments of repeated use of filtrate from a COE epoxidation run using Au/SiO$_2$-A as Au source. Run 1 (panel I) was a reaction similar to those in Fig. 1. Afterwards, the mixture was hot-filtered. 2 ml of filtrate from Run 1 was used for Run 2 by adding COE and an internal standard of decane to 10.8 ml total volume (panel II), and 2 ml of the reaction mixture of Run 2 was mixed with fresh COE/decane for Run 3 (panel III).

detected during the induction period but was prominent when the catalytic activity was high (Supplementary Fig. 11). This suggests that they are related to the catalytic activity. Also consistent with their catalytic relevance was the observation that, in general, solutions that showed high activities for COE epoxidation also exhibited highly intense peaks (Supplementary Fig. 10).

The emission peaks of Au/SiO$_2$-A derived filtrate shifted to ~425 and 375 nm upon 12- and 120-fold dilution with ethanol, respectively, and the corresponding excitation peak maxima to ~360 and 310 nm (Fig. 5a, spectra b and c). Their intensities decreased with increasing dilution. Blue-shifts of peaks by about the same amount were also observed when diluted with a non-coordinating, nonpolar solvent like COE (Supplementary Fig. 12). Likewise, dilution of filtrates derived from AuCl with ethanol also resulted in peak shifts to shorter wavelengths (Supplementary Fig. 9a). The similar extents of blue-shift with dilution with ethanol and COE suggested that it was not due to solvent effect. Instead it was due to changes in the Au cluster size in response to the changed Au and ligand concentrations of the solution.

The broad spectral bandwidth of the filtrate emission peaks (Fig. 5a, curve a) suggests that they are composites of overlapping emission peaks from Au clusters of different sizes. This was confirmed by the interesting dependence of intensity with excitation wavelength for solutions of different Au concentrations as is shown in Fig. 5b for Au/SiO$_2$-A derived filtrate and Supplementary Fig. 9b for AuCl derived filtrate. Figure 5b compares the emission spectra obtained with 400 and 360 nm excitation of a filtrate and its 12-fold diluted solution. With 400 nm excitation, the peak intensity of the filtrate was higher than the 12-fold diluted sample. With 360 nm light excitation, the reverse was observed—the emission peak of the 12-fold diluted sample was more intense than the undiluted filtrate. Correspondingly, the absorption at 360 nm was more intense for the diluted sample (Fig. 5b, inset). The reversal in emission peak intensities for the original and diluted filtrates upon switching from 400 to 360 nm excitation can be interpreted to be due to a larger population of smaller clusters in the diluted sample, and smaller Au clusters are known to be more efficient in photoemission at shorter wavelengths[21]. The solubilized Au species are in equilibrium with each other in the solution, including those not detected by fluorescence but clearly discernable in aberration-corrected TEM images (for example, Au atoms). Dilution decreased the concentrations of both Au and coordinating ligands, and affected the equilibrium size distribution of Au clusters.

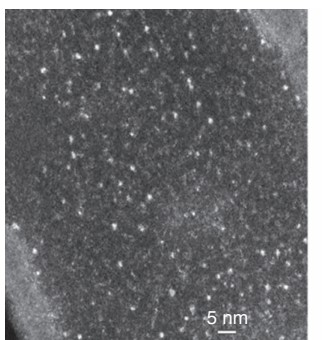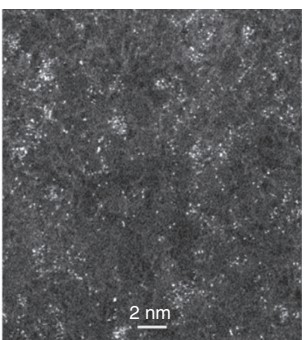

**Figure 4 | Aberration-corrected electron microscopy images of filtrates.** TEM images of filtrate from Au/SiO$_2$-A collected after conversion reached 18%. The filtrate was diluted with ethanol and dried on the grid. Scale bar, 5 nm and 2 nm for the left and right images.

**Extension to other systems**. It is possible that the function of solubilized Au to inject initiator/propagators into the auto-oxidation reaction can be extended to other olefins, and be replaced by other metals. Both of these were tested. PtCl$_4$ was tested using the same procedure as for AuCl, and the results are show in Supplementary Fig. 13. The reaction profile was very similar to using Au; there was an induction period after which effective, selective epoxidation occurred. At the end of the reaction, the mixture was filtered hot, and the filtrate was used to initiate COE epoxidation without an induction period (Supplementary Fig. 13). The filtrate was found to have 0.1 mg ml$^{-1}$ of solubilized Pt, and it exhibited an emission spectrum (Supplementary Fig. 14). Thus, solubilized Pt can function like Au.

The filtrate derived from an Au-catalysed COE reaction and with hydroperoxide removed by PPh$_3$ was tested for cyclohexene oxidation at 60 °C using molecular O$_2$. After 6 h, ~20% of

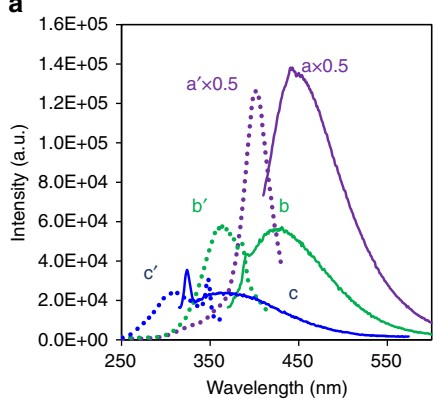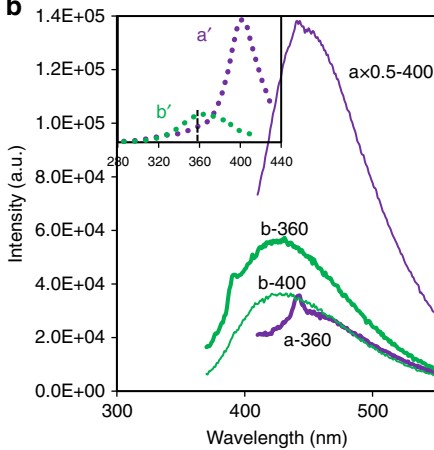

**Figure 5 | Ultraviolet–visible excitation and emission spectra.** (**a**) Emission (solid lines) and excitation (dotted lines) spectra of: (a,a′) filtrate derived from Au/SiO$_2$-A (scaled by 0.5); (b,b′) 12-fold ethanol-diluted filtrate; and (c,c′) 120-fold ethanol diluted filtrate. The parent filtrate contained 108 ng ml$^{-1}$ of Au. Emission spectra a–c were collected using $\lambda_{excitation} = 405$, 360, and 310 nm, respectively. Excitation spectra a′–c′ were collected by monitoring at $\lambda_{emission} = 450$, 425 and, 375 nm, respectively. The sharp features in spectra c,c′ are instrument artefacts (Supplementary Fig. 15). (**b**) Emission spectra of filtrate (a) and 12-times diluted filtrate (b) collected at $\lambda_{excitation} = 400$ (−400 curves) and 360 nm (−360 curves). Inset showed the excitation spectra of the filtrates.

cyclohexene was oxidized and the major product was cyclohexene hydroperoixde (~60%), with minor amounts of 2-cyclohexen-1-ol, 2-cyclohexen-1-none, and cyclohexane epoxide, a distribution similar to that from auto-oxidation[22,23]. Without the Au-derived filtrate, there was no reaction. Thus, the solubilized Au was capable of initiating oxidation of other cyclic alkenes.

## Discussion

We have demonstrated definitively that solubilized Au, derived from the Au nanoparticles on a support or Au salts, is the catalytic relevant species in COE epoxidation. The Au nanoparticles and Au salts are precatalysts and their roles comprised of serving as a source of Au as well as the producer of a minute number of stabilizing ligands for the solubilized Au at the earliest phase of the induction period. The formation rate of the initial solubilized Au clusters depends on the ease of dissolution of Au from its source and the ability of the Au source to catalyse the oxidation of COE into effective chelating oxygenated hydrocarbons. Thus, the induction period can be shortened by the addition of effective chelates, such as diols and molecules with multiple functional groups.

Figure 6 shows the manner in which the solubilized Au atom clusters interface with the auto-oxidation cycle (which is represented by a generic cycle since its details are not important for the present discussion). Au accelerates COE oxidation in two ways. In the initial phase, they play the essential role of producing initiators for the auto-oxidation phase of the COE reaction, which in turn generates more oxygenated ligands to form additional atomic clusters. The high activity of Au clusters in this phase was determined in filtrates with the organic hydroperoxide removed by $PPh_3$ to be equivalent to a TOF of $440 s^{-1}$. In the auto-oxidation phase, these solubilized Au clusters continue to generate additional initiators to accelerate the auto-oxidation cycle. By comparing the rates of this phase of the reaction in experiments where a filtrate was used repeatedly through multiple cycles of dilution and thus decreasing Au contribution, the contribution of Au to the TOF at 50% conversion was $92 s^{-1}$. The ability of solubilized Au to continuously generate initiators distinguishes it from free radical initiators such as $t$-butyl hydroperoxide that are consumed in the initiation process. In this phase, the data also showed that the solubilized Au clusters do not participate in the auto-oxidation cycle directly, that is, Au is not a necessary component for auto-oxidation. It should be noted that if this point was not recognized and all the epoxidation activity was attributed to solubilized Au, the TOF would mistakenly become ~1,500 $s^{-1}$ for this set of data. Since the auto-oxidation was rather insensitive to the concentration of Au (within a factor of two for [Au] ranging from 20 to 120 ng ml$^{-1}$),

an inflated value of TOF could reach as high as 7,000 $s^{-1}$. For reactions that involve complex radical chain mechanism such as cyclohexane oxidation, often the contributions to the rate by Au catalyst and auto-oxidation are not separated but attributed entirely to Au, resulting in over-estimated values of TOFs[24,25].

This understanding of the role of solubilized Au resolves the controversy about the nature of the active form of supported Au catalysts for this reaction. In fact, such controversy can be found also for other Au-catalysed liquid phase oxidation reactions. For example, Au/C catalysts with large Au particles are active for selective hydrocarbon oxidation[17], whereas small Au atom clusters are reported to be the active form for thiophenol oxidation[26] and stilbene epoxidation[27]. An induction period is common for these reactions, yet few conducted hot filtration and other characterization to determine if a phenomenon similar to those observed here may occur. Thus further investigation is warranted.

The ability of Au to generate initiator is applicable to other inorganic initiators. Indeed, we showed that this can be accomplished with solubilized Pt formed from PtCl$_4$. The amount of solubilized metal atom clusters (of the order of ng ml$^{-1}$) needed as catalyst initiators is significantly less than traditional catalysts employed for this class of reactions. Consequently, the very low costs associated with catalyst material and oxidant (O$_2$) enhance the appeal of this process for further development. From another angle, the fact that solubilized Au is effective in initiating other selective oxidation reactions such as cyclohexene oxidation with O$_2$ broadens its potential field of application. There is a high probability that it is also effective in initiating other hydrocarbon oxidation reactions that are susceptible to free radical initiation, such as cyclohexane. The transferability of the formation and catalytic role of solubilized metal atom clusters to other systems offers exciting opportunities.

The solubilized Au is present as a mixture of species ranging from Au atoms to clusters of ~0.7 nm. The most abundant ones at the end of a reaction are 7–8 atom clusters judging from the positions (~450 nm) and intensities of the emission peaks. The optical spectra of these clusters are absent during the induction period but are intense at the high-activity phase of the reaction, suggesting their catalytic role. In line with their catalytic role is that in different reaction solutions, the intensity of the 450 nm emission peak trended with the rates of COE conversions.

## Methods

**General information.** Except otherwise noted, all chemicals were purchased and used without purification. Additional details of their sources and experimental procedures are in the Supplementary Section 1 Methods and Material. When applied, the stabilizer in COE was removed by distillation.

**Catalysts.** Au/SiO$_2$-A (2 nm average Au particle size) was prepared using Au(en)$_2$Cl$_3$ precursor, which was prepared by adding ethylenediamine (en, en/Au = 2.65) to a water/ethanol solution of HAuCl$_4$.3H$_2$O, similar to the procedure of Zhu *et al.*[28]. After filtering, enough of this filtrate at 4.2 mM was added to fume silica at 40 °C for a nominal Au loading of 2.2 wt.%. The pH of the solution was adjusted to 9 by dropwise addition of 0.75M en solution. The mixture was stirred for 2 h at room temperature, filtered, washed first with water at room temperature, and then with 40 °C water. After calcination in a flowing O$_2$/O$_3$ mixture at 150 °C. the resulting powder was light yellow in colour, and had a Au loading of 1.2 wt.%

Au/SiO$_2$-B1 (average 0.9 nm Au particles) and –B2 (4.5 nm), in which the Au particles were partially encapsulated by thin islands of SiO$_2$, were synthesized similar to our previously published method[29]. Briefly, poly(methylhydro)siloxane with trimethylsiloxy termination (PMHS) was partially oxidized (~50%) with water over Pd/ carbon catalysts. The resulting silanol groups were further reacted with N-methyl-aza-2,2,4-trimethylsilacyclopentane to form amine functionalized PMHS. Then Au(THT)Cl was added to the amine functionalized PMHS (amine/Au = 10) and stirred for 0.5 h before the introduction of fumed silica. The mixture was filtered and washed, and the solid was calcined in a flowing O$_2$/O$_3$ mixture at 150 °C. The final Au weight loading of the sample was 0.9 wt.%. The two samples differed in the degree of washing and dryness before calcination.

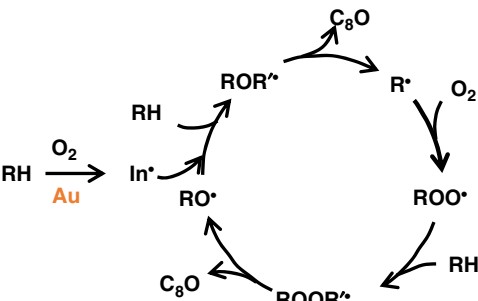

**Figure 6 | Proposed function of Au in cyclooctene epoxidation with O$_2$.** Proposed catalytic function of Au atomic clusters in the autooxidation cycle for the formation of cyclooctene epoxide (C$_8$O). R = c-C$_8$H$_{13}$, R' = c-C$_8$H$_{14}$, and In• is free radical initiator. Note that Au functions to generate free radical initiators and is not necessary in the C$_8$O formation cycle.

**Characterization.** Emission and excitation spectra of the filtrates were recorded using Photon Technology International Model QM-2. An emission spectrum was collected by first determining the wavelength of the excitation light that would yield the maximum emission intensity. Once determined, the emission spectrum was collected using this excitation wavelength. Excitation spectrum was generated by monitoring the fluorescence emission at the wavelength of the emission maximum while varying the excitation wavelength. High-performance liquid chromatography grade tetrahydrofuran (THF) or ethanol was used for sample dilution when used. Inductively coupled plasma mass spectrometry was performed with ThermoiCAP Q Inductively Coupled Plasma Mass Spectrometry. Water was used to determine instrument artefacts (Supplementary Table 7), and background emissions due to dissolved organics and solvents were also measured (Supplementary Fig. 15). For preparation, 1 ml of the sample was heated to 850 °C to remove all organics. The remaining solid was dissolved in concentrated $HNO_3/HCl$, and diluted with water. High-angle annular dark-field scanning transmission electron microscopy images were obtained on a JEOL ARM-200F aberration-corrected STEM (AC-STEM) operated at 200 kV with a nominal spatial resolution of 0.08 nm in the STEM mode. The TEM/STEM samples were prepared by dipping the lacey carbon covered copper TEM grid directly into ethanol-diluted filtrate solutions. After drying, the samples formed a thin film that covered the holes of the lacey carbon film.

**Catalytic tests.** Because of Au plating and oxidation activity of residual reaction liquids, it was essential that the reactor vessel was thoroughly cleaned after each use. An elaborate procedure as described in Supplementary Information was employed (Supplementary Methods Section 1.4.4), such that without adding a Au source, there was no reaction of COE for at least 10 h. The catalytic oxidation of COE was conducted in the absence of light in a 50 ml three-necked round bottom flask equipped with a fine grit glass gas disperser for bubbling $O_2$ through the liquid and a condenser maintained at −5 °C (Supplementary Fig. 16). In a typical reaction, 10 ml of COE, 1 ml n-decane (internal standard), Au source and a Teflon-coated magnetic stirrer were loaded into the flask. The mixture was sonicated for 10 min in the dark, and then placed in an oil bath preheated to 100 °C with stirring. After the mixture reached 100 °C, the reaction was commenced by starting the flow of ultrahigh purity $O_2$. 0.1–0.2 ml aliquots of the reaction mixture were removed with a syringe at different time intervals, diluted with anhydrous THF and frozen until analysis. At completion of the run, the mixture was filtered immediately using a 0.2 µm syringe filter. GC analysis was with a Agilent 6890 GC equipped with a FID and a Agilent J and W DB-624 capillary column (30 m × 0.25 mm × 0.25 µm). GC-MS spectrometry (Agilent GC-7890 A, MS-5975). and $^1H$ and $^{13}C$ NMR were used to assist product identification. In particular, cyclooctene hydroperoxide was detected using $^1H$ NMR (Supplementary Fig. 17). GC sensitivity factors of the different products were calibrated with commercially available standards when possible. Cyclooctanol and cyclooctanone were used as substitutes for cyclooctenol and cyclooctenone.

Conversion and selectivity data were calculated using GC area ratios of molecules of interest to the internal standard decane, after correction for evaporative loss of COE. The latter was estimated from measurements made with cyclooctane (Supplementary Fig. 2), which has a similar boiling point as COE (Supplementary Table 8) and is non-reactive in air, and taking into account decreasing mole fraction with conversion. Procedural details and equations used for correction for evaporation, conversion and selectivity are in Supplementary Methods Section 1.4.3.

**Titration for peroxides.** Peroxide test paper (0.5–25 p.p.m. range) was used to test for total destruction of peroxide during titration with triphenylphosphine (5 uM in dichloromethane or ethanol). Since peroxide test paper may not be sensitive to polymeric hydroperoxide, KI and starch test was also used to verify the absence of polymeric hydroperoxide.

**Data availability.** All data are available from the authors upon reasonable request.

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

## Acknowledgements

The authors acknowledge support of this work by the US Department of Energy, Office of Science, Office of Basic Energy Sciences under Award Number DOE DE-FG02-03-ER15457, Y.Y. Wu for a sample of Au/SiO2, and Anyang Peng for experimental assistance, China Scholarship Council for support of L.Q. and Z.W., CAPES Foundation (Brazil) for support of H.J.d.S., the National Science Foundation (CHE—1465057) for support of STEM work (J.L.), and use of facilities in the John M. Cowley Center for High Resolution Electron Microscopy at Arizona State University.

## Author contributions

Z.W. and L.Q. collected most of the data and participated in interpretation of data and planning of experiments. E.V.B. performed NMR measurements and helped interpretation of all data, J.L. collected all aberration correction TEM images, H.J.d.S. prepared some of the catalysts and collected data with them, T.L. helped supervise Z.W., and M.d.C.R. helped supervised H.J.d.S., M.C.K. and H.H.K. provided overall supervision and set direction of the project.

## Additional information

**Competing interests:** The authors declare no competing financial interests.

DOI: 10.1038/ncomms15669    OPEN

# Erratum: Stable and solubilized active Au atom clusters for selective epoxidation of *cis*-cyclooctene with molecular oxygen

Linping Qian, Zhen Wang, Evgeny V. Beletskiy, Jingyue Liu, Haroldo J. dos Santos, Tiehu Li, Maria do C. Rangel, Mayfair C. Kung & Harold H. Kung

*Nature Communications* 8:14881 doi: 10.1038/ncomms14881 (2017); Published 28 Mar 2017; Updated 11 Aug 2017

The original version of this Article contained an error in which the second affiliation 'Chemical and Biological Engineering Department, Northwestern University, Evanston, Illinois 60208, USA'; was inadvertently switched with the third affiliation 'School of Materials Science and Engineering, Northwestern Polytechnical University, Xi'an, Shaanxi 710072, China'.

Additionally, 'Shaanxi' was misspelt as 'Shaansi'. This has now been corrected in both the PDF and HTML versions of the Article.

