## [Peer Review File · Nature Communications]

Reviewers' comments:

Reviewer #1 (Remarks to the Author):

This paper presents a study of the epoxidation of cyclo octene catalysed by gold. The paper proposes that soluble gold species stabilised by organic ligands derived from the oxidation of cyclooctene are responsible for the epoxidation. This is a very novel proposal; indeed an extraordinary proposal and as such extraordinary evidence is required to support it. At present I am not sure this is the case and I will set out the questions that I feel need to be resolved. The evidence is that there is an induction period and that during this time the soluble gold species are formed. However, the induction period has been well reported for this reaction with gold catalysts before and this has been explained as being the period of time during which the reactive epoxidising species are formed from the oxidation of the initial allylic oxidation products. Like the soluble gold species these are soluble and would survive a hot filtration. The time on line product distribution in the literature supports this explanation of the induction period. In reality the origin of the induction period is the key evidence supporting this new proposal. Yet there can be two explanations (a) the soluble gold species which act as the epoxidation catalyst and (b) the formation of epoxidising species from the allylic reaction products. In the present paper I do not see the clear evidence that says (a) is correct and (b) is wrong. This evidence is needed. For example does the concentration of Au species match the epoxidation activity in some way that the formation of the allylic products does not. This is a difficult conundrum and the paper is clearly interesting but I don't think in its present form it makes the case fully.

Reviewer #2 (Remarks to the Author):

Referees report on "Stable, Solubilized Active Au Atom Clusters for Selective Epoxidation of Cis-Cyclooctene with Molecular Oxygen" submitted to Nature Communications by Linping Qian, Zhen Wang, Evgeny V. Beletskiy, Jingyue Liu, Haroldo J. dos Santos, Tiehu Li, Maria do C. Rangel, Mayfair C. Kung,* Harold H. Kung

The authors present a study of gold catalyzed epoxidation of a cyclic alkene, in which they convincingly demonstrate catalysis likely results from solubilized gold species even when starting with a solid catalyst and accounting for the high background rate of oxidation due to peroxide / hydrocarbon species. The reactions are sparged with oxygen and the auto-oxidation kinetics are similar to those seen by Hutchings et al. or indeed the authors using a radical initiator, indicating that although catalysis occurs at a higher rate when the gold species are available, they serve only as an alternative radical initiator / generator for the process.

The idea that small gold particles should behave in this way for oxidations has been reported previously by Corma and coworkers [Corma, A.; Concepción, P.; Boronat, M.; Sabater, M. J.; Navas, J.; Yacaman, M. J.; Larios, E.; Posadas, A.; López-Quintela, M. A.; Buceta, D.; Mendoza, E.; Guilera, G.; Mayoral, A., Exceptional oxidation activity with size-controlled supported gold clusters of low atomicity. *Nat Chem* 2013, 5 (9), 775] – so although not for epoxidation the idea they can act in this way is not novel, the present manuscript is only a specific example of this process. Interestingly small clusters of gold have been also studied theoretically [Pal, R.; Wang, L.-M.; Pei, Y.; Wang, L.-S.; Zeng, X. C., Unraveling the Mechanisms of O₂ Activation by Size-Selected Gold Clusters: Transition from Superoxo to Peroxo Chemisorption. *J. Am. Chem. Soc.* 2012, 134 (22), 9438.], and with the clusters supported on solid supports, shown for both Au and Ag metals in this type of reaction [Lee, S.; Molina, L. M.; López, M. J.; Alonso, J. A.; Hammer, B.; Lee, B.; Seifert, S.; Winans, R. E.; Elam, J. W.; Pellin, M. J.; Vajda, S., Selective Propene Epoxidation on Immobilized Au_{6–10} Clusters: The Effect of Hydrogen and Water on Activity and Selectivity. *Angew. Chem.* 2009, 121 (8), 1495; Lei, Y.; Mehmood, F.; Lee, S.; Greeley, J.; Lee, B.; Seifert, S.; Winans, R. E.; Elam, J. W.; Meyer, R. J.; Redfern, P. C.; Teschner, D.; Schlögl, R.; Pellin, M. J.; Curtiss, L. A.; Vajda, S., Increased Silver Activity for Direct Propylene Epoxidation via Subnanometer Size Effects. *Science* 2010, 328 (5975), 224.].

The authors report that the process occurring is selective epoxidation, but they conduct the reaction in the solvent, and calculate selectivity as a fraction of liquid products seen, ignoring the pathway to complete oxidation giving CO₂ that often occurs after formation of carbonyl (unwanted) side products. Absence of other products is not confirmed by a mass balance (see below).

While it is an interesting report in this area, and is of likely interest to the readers of more specialist journals in the field of catalysis, I do not believe the work represents a significant new step forward that justifies publication in Nature Communications. The work is generally carefully and thoroughly reported and possible for another to reproduce with the level of detail provided by the methods. Some minor comments of clarification are below, but addressing them does not alter this view.

- Typo in abstract – ‘understanding’ should read ‘understandings’
- In references for Au catalysed oxidation, other important examples include those above and Gajan, D.; Guillois, K.; Delichère, P.; Basset, J.-M.; Candy, J.-P.; Caps, V.; Coperet, C.; Lesage, A.; Emsley, L., Gold Nanoparticles Supported on Passivated Silica: Access to an Efficient Aerobic Epoxidation Catalyst and the Intrinsic Oxidation Activity of Gold. *J. Am. Chem. Soc.* 2009, 131 (41), 14667.
- For the “initial rates” quoted – over what period are these obtained / given the difficulty of identifying differences visually in the rate data, are the errors (mostly around +/- 0.2 mmol h⁻¹) derived from the analysis result of repeats of reactions?
- The authors detail detection of around the detection of 7-8 atom clusters, and say their detection with the onset of reactivity demonstrates catalytic relevance. While it isn’t clear what “relevance” means exactly, care is needed here as their production could just be coincident with sufficient leaching in solution, or smaller/larger catalytically competent minority species – their observation doesn’t necessarily affirm they are the catalyst in a system as complex as this – although this conclusion would concur with other references above as being in the appropriate size range.
- I’m confused by the fact the internal standard is used to verify reactant loss, but no attempt is made to look at yield with this number based on the products that are actually observed – i.e. a mass balance (after allowing for the COE loss). It must be possible to work out from the internal standard how much of each product is made, without relying on reactant conversion and selectivity to do the calculation (removing the problem of differential loss of reactants) – if they are detected surely the yield can be calculated from their known amounts directly? This would also give a much clearer idea whether CO₂ is being formed or not – presenting this evidence may help show whether or not the reaction is really selective to epoxide if it is known approximately how much COE is consumed and how much epoxide is produced.

Reply to reviewers' comments, re manuscript: "Stable, Solubilized Active Au Atom Clusters for Selective Epoxidation of *Cis*-Cyclooctene with Molecular Oxygen" by Linping Qian, et al.

Reviewer #1:

Comments:

1. "..... This is a very novel proposal; indeed an extraordinary proposal and as such extraordinary evidence is required to support it. At present I am not sure this is the case and I will set out the questions that I feel need to be resolved. The evidence is that there is an induction period and that during this time the soluble gold species are formed. However, the induction period has been well reported for this reaction with gold catalysts before and this has been explained as being the period of time during which the reactive epoxidising species are formed from the oxidation of the initial allylic oxidation products. Like the soluble gold species these are soluble and would survive a hot filtration."
2. ".....In reality the origin of the induction period is the key evidence supporting this new proposal. Yet there can be two explanations (a) the soluble gold species which act as the epoxidation catalyst and (b) the formation of epoxidising species from the allylic reaction products. In the present paper I do not see the clear evidence that says (a) is correct and (b) is wrong. This evidence is needed. For example does the concentration of Au species match the epoxidation activity in some way that the formation of the allylic products does not. This is a difficult conundrum and the paper is clearly interesting but I don't think in its present form it make the case fully."

Reply:

The concern of the reviewer is that strong evidence is needed to show that the induction period is due to formation of soluble gold species which act as the epoxidation catalyst (point 2) and NOT the formation of epoxidizing species from the allylic reaction products (point 1).

Since these two points are strongly related, we will attempt to answer both simultaneously. The main results of the experiments below are included in the revised manuscript as three new, highlighted paragraphs.

Following the suggestion of the reviewer, we conducted additional experiments to collect data to support what we believe are the two novel ideas presented in this paper. These novel ideas are: (a) the catalytic function of solubilized Au is to **inject initiators** for the free-radical auto-oxidation reaction which can proceed without a catalyst, and (b) the active Au catalyst, which consists of clusters of 6-8 atoms, is formed and stabilized by the oxidation products.

We believe that the first idea is novel, since it has not been suggested before. It is illustrated by the scheme shown in Figure Re-1 (also added as Fig. 6 in manuscript text). It is a new concept, since a typical catalyst participates in the formation of every product molecule. In this new role, the product (cyclooctene epoxide C₈O) is formed without a Au catalyst by the free radical auto-oxidation reaction cycle. However, the auto-oxidation needs to be initiated by an initiator, and Au provides the initiator by catalytic oxidation of cyclooctene. Thus, when an Au pre-catalyst is used in the first reaction, an induction period appears to form the active Au atomic clusters. But when the active Au catalyst is already present as in the filtrate-initiated reactions, there is no

induction period even when all hydroperoxide is removed with triphenylphosphine. In addition, because this function of Au is catalytic, it continues to inject initiators into auto-oxidation reaction cycle, and enhances the reaction rate throughout the reaction. This differs from other free-radical initiators such as tert-butyl hydroperoxide which is consumed in the stoichiometric formation of initiator. The latter initiators are useful only once.

Figure Re-1 (also as Fig. 6 in manuscript). Proposed catalytic function of Au atomic clusters in the autooxidation cycle for the formation of cyclooctene epoxide (C₈O). R = c-C₈H₁₇, R' = c-C₈H₁₆, and In• is free radical initiator. Note that Au functions to generate free radical initiators and is not needed in the C₈O formation cycle.

The critical experiment to substantiate this model is to demonstrate that under otherwise identical conditions and in the absence of hydroperoxide, there is a substantial induction period without Au, but no induction period when Au is present. Since reaction rates beyond about 15-20% conversion in a batch reactor are dominated by the auto-oxidation reaction, the most useful and definitive data are collected as initial rates. The results of this set of experiments are shown in Table Re-1 (also added as Supplementary Table S-6).

These experiments were conducted as follows. Experiment 1 was initiated by using 2 mL of the stock Au filtrate together with 1.0 equivalent (with respect to detected hydroperoxide) of triphenylphosphine (PPh₃) and a fresh batch of COE. A comparative experiment was performed in which the *t*-BuOOH stock control was used together with 1.0 equivalent PPh₃ (Exp. 3).

As observed previously (Fig. 2 in manuscript), Exp. 1 showed no induction period. Its initial COE consumption rate was 2.1 mmol h⁻¹ as shown in the Table. In contrast, using the *t*-BuOOH filtrate with added PPh₃ in Exp. 3, there was a significant induction period, and the initial reaction rate was not measurable. The result of Exp. 3 clearly showed that PPh₃ was very effective in destroying the initiators generated by *t*-BuOOH and the auto-oxidation reaction. Thus, the initial reaction observed in Exp. 1 is attributed to the presence of solubilized Au. That is, Au generates initiators for the reaction. These results confirmed what was reported in the original manuscript (Fig. 2), where a similar comparison between stock Au filtrate and *t*-BuOOH filtrate was made using 1.05 eq. PPh₃.

Table Re-1 (also as Supplementary Table S-6 in text). Effects of solubilized Au concentration and triphenylphosphine addition on the initial cyclooctene consumption rate.

Exp.	Sample ^a	Initial rate, ^b mmol h ⁻¹
1	5.4 times diluted stock Au filtrate + 1.0 eq. PPh ₃	2.1±0.3
2 ^c	157 times diluted stock Au filtrate + 1.0 eq. PPh ₃	0.10±0.05
3	5.4 times diluted stock t -BuOOH filtrate + 1.0 eq. PPh ₃	~0.00

^a Stock Au filtrate was

derived from a reaction using Au/SiO₂-A, and contained ~100 ng mL⁻¹ solubilized Au. Stock *t*-BuOOH was derived from a reaction using *tert*-butyl peroxide as initiator. An amount of PPh₃, 1.0 equivalent of hydroperoxide present in the reaction, was added to each of these solutions prior to the experiment. ^b Initial cyclooctene consumption rate. ^c The procedure for experiment 3 is shown in the diagram below. 2 mL of the reaction mixture at the end of Exp. 1 was used to initiate a second reaction. Then 2 mL of the product mixture of this second reaction was mixed with 1.0 eq. PPh₃ to initiate Exp. 2.

An additional experiment was conducted to probe the effect of solubilized Au concentration on the initiation rate (Exp. 2 in Supplementary Table Re-1). Starting with a stock Au filtrate, a series of three consecutive runs was performed in which 2 mL aliquot of the filtrate of the preceding run was used as initiator. Thus, at the start of each run in the series, the organic reactive intermediate would be relatively similar, but the solubilized Au would be repeatedly diluted, such that its concentration in Exp. 1 was 5.4 times more concentrated than the next run, and 29 times more concentrated than Exp. 2 (which was 157 times more dilute than the stock filtrate). In addition, 1 eq. of PPh₃ corresponding to the hydroperoxide present in the aliquot was added to Exps. 1 and 2 at the beginning of the runs. The initial rate observed for Exp. 2 decreased by ~25 times compared with Exp. 1 (Supplementary Table S-6).

Summarizing, these data together with similar ones in the original manuscript showed the following:

- In mixtures containing the same equivalent of PPh₃, i.e. Exp. 1 versus 3, the mixture containing Au was always more reactive than without Au.
- Comparing mixtures containing different concentrations of Au and same equivalent of PPh₃ (Exp. 1 versus 2), the mixture with a higher Au concentration was more reactive.
- PPh₃ was very effective in destroying reactive intermediates responsible for free radical auto-oxidation chain reactions (Exp. 3).

Reviewer #2:

Comment 1: The authors present a study of gold catalyzed epoxidation of a cyclic alkene, in which they convincingly demonstrate catalysis likely results from solubilized gold species even when starting with a solid catalyst and accounting for the high background rate of oxidation due to peroxide / hydrocarbon species. The reactions are sparged with oxygen and the auto-oxidation kinetics are similar to those seen by Hutchings et al. or indeed the authors using a radical initiator, indicating that although catalysis occurs at a higher rate when the gold species are available, they serve only as an alternative radical initiator / generator for the process.

Response: We are delighted that the reviewer recognized the new concept and contribution of our work.

Comment 2: The idea that small gold particles should behave in this way for oxidations has been reported previously by Corma and coworkers [*Ref. a*, Nat Chem 2013, 5 (9), 775] – so although not for epoxidation the idea they can act in this way is not novel, the present manuscript is only a specific example of this process. Interestingly small clusters of gold have been also studied theoretically [*Ref. b*, J. Am. Chem. Soc. 2012, 134 (22), 9438.], and with the clusters supported on solid supports, shown for both Au and Ag metals in this type of reaction [*Ref. c*, Angew. Chem. 2009, 121 (8), 1495; *Ref. d*, Science 2010, 328 (5975), 224.].

Response:

We are familiar with these publications, but chose not to include them in the original manuscript because we focused on liquid phase epoxidation reactions, especially using molecular oxygen, and highlighted the uncertainties in the literature of the nature of the active Au species for those reactions.

These references mentioned by the reviewer were not included in the original manuscript because of the following reasons:

- a) Corma's work is not on epoxidation.
- b) Neither Corma's work on disulfide oxidation nor Gajan's work on stilbene epoxidation mentioned hot filtration. Thus it is unclear whether they have definitively excluded contributions from solubilized Au.
- c) The Au clusters in these reports are unstable (more details below), whereas ours are highly stable.
- d) The Au clusters in these reports participate in the catalytic formation of the products directly. This is not the case in our system. The solubilized Au clusters in our work do not catalyze the product formation directly. Instead, they catalyze the initiation of the reaction to form the products. This is a new and unusual function of a catalyst that has not been reported before. Importantly, in this new role, the Au catalyst is NOT consumed, whereas other traditional free radical initiators are consumed stoichiometrically.

Regarding point (c) on catalyst stability, in *Reference a*, the catalyst was inactive during the 2 min induction, and then after activation, at 12 min the activity was significantly curtailed due to Au agglomeration. Thus, it is much less clear than our system what is the actual active form of Au. Similarly, *Ref. d* reported catalyst instability. The Ag₃ clusters abruptly transition to

nanoparticles at 110°C, although the temperature programmed reaction profile did not reflect this abrupt structural change in either the activity or the selectivity. Thus, again, the exact active form is not clear. *References b, c and d* dealt with *model systems* of clusters deposited on flat surfaces for gas phase reactions. Thus, a priori, it is unknown whether conclusions from those systems can apply to liquid phase, solubilized systems and free radical reactions. Indeed, it would be impossible to deduce from those studies our new finding of Au catalyzing initiation of free radical reactions. Finally, the computational studies included in *Reference b* were related to oxygen adsorption on anionic Au clusters. The results have to be extrapolated to catalysis.

Nonetheless, we added Corma's (*Ref. a*) and Gajan's work (referred to by the reviewer in a later comment) to the revised manuscript in the third paragraph of Conclusion.

Comment 3.

The authors report that the process occurring is selective epoxidation, but they conduct the reaction in the solvent, and calculate selectivity as a fraction of liquid products seen, ignoring the pathway to complete oxidation giving CO₂ that often occurs after formation of carbonyl (unwanted) side products. Absence of other products is not confirmed by a mass balance (see below).

- I'm confused by the fact the internal standard is used to verify reactant loss, but no attempt is made to look at yield with this number based on the products that are actually observed – i.e. a mass balance (after allowing for the COE loss). It must be possible to work out from the internal standard how much of each product is made, without relying on reactant conversion and selectivity to do the calculation (removing the problem of differential loss of reactants) – if they are detected surely the yield can be calculated from their known amounts directly? This would also give a much clearer idea whether CO₂ is being formed or not – presenting this evidence may help show whether or not the reaction is really selective to epoxide if it is known approximately how much COE is consumed and how much epoxide is produced.

Reply:

We have specifically addressed the question of material balance in the original manuscript. The reactions were conducted without use of solvent. Using the internal standard (decane, chosen for its low vapor pressure and negligible loss under our conditions) to carefully account for evaporative loss of reactant cyclooctene throughout the reaction, we were able to obtain carbon balances of ±10% or better over the entire course of reaction, some as long as 20+ hours, and as low as ±1% at low conversions. These values were reported in the original manuscript both in the main text (second paragraph of Results) and in Supporting Information (Footnote of Table S-3). The good carbon balance must mean that we are able to account for all major products. That is, complete oxidation to CO₂ (or CO) must be a very minor contribution to the overall reaction.

We are confused by the reviewer's statement: "I'm confused by the fact the internal standard is used to verify reactant loss, but no attempt is made to look at yield with this number based on the products that are actually observed – i.e. a mass balance (after allowing for the COE loss)." We reported carbon balance, conversions based on initial cyclooctene present (NOT conversion based on products detected), selectivity, and yield all based on initial cyclooctene present. The equations to calculate these values are shown in Supporting Information (Eq. 1 to 5). They are textbook equations. By definition, yield is product of conversion and selectivity. Thus, we have

provided all the data that the reviewer asked for in exactly the form that he/she said should be done. Again, using these values, we concluded that the carbon balance was good and complete oxidation that the reviewer is concerned with is not significant. In order to emphasize this, we added a statement in the caption of Figure 1 that “Conversion is defined based on initial COE present.”

Comment 4:

While it is an interesting report in this area, and is of likely interest to the readers of more specialist journals in the field of catalysis, I do not believe the work represents a significant new step forward that justifies publication in Nature Communications. The work is generally carefully and thoroughly reported and possible for another to reproduce with the level of detail provided by the methods. Some minor comments of clarification are below, but addressing them does not alter this view.

Reply:

We respectfully disagree with the reviewer. As detailed in reply to comment 2, we believe that we have identified a new catalytic function in selective oxidation catalysis that has not been recognized before, as shown in Figure Re-1 in our reply to Reviewer 1’s comment. This is an unconventional function of catalysis. In addition, the discovery that solubilized Pt clusters can be generated and function in a similar manner is unexpected and has not been reported before.

Other minor comments:

- Typo in abstract – ‘understanding’ should read ‘understandings’

Reply: This is corrected.

- In references for Au catalysed oxidation, other important examples include those above and Gajan, D.; Guillois, K.; Delichère, P.; Basset, J.-M.; Candy, J.-P.; Caps, V.; Coperet, C.; Lesage, A.; Emsley, L., Gold Nanoparticles Supported on Passivated Silica: Access to an Efficient Aerobic Epoxidation Catalyst and the Intrinsic Oxidation Activity of Gold. *J. Am. Chem. Soc.* 2009, 131 (41), 14667.

Reply: Reference added.

- For the “initial rates” quoted – over what period are these obtained / given the difficulty of identifying differences visually in the rate data, are the errors (mostly around +/- 0.2 mmol h⁻¹) derived from the analysis result of repeats of reactions?

Reply:

The initial rates were derived from data collected in the first 2 to 3 hours, emphasizing the first hour. The uncertainties were estimated from a combination of repeat experiments and estimation within each experiment.

- The authors detail detection of around the detection of 7-8 atom clusters, and say their detection with the onset of reactivity demonstrates catalytic relevance. While it isn’t clear what

“relevance” means exactly, care is needed here as their production could just be coincident with sufficient leaching in solution, or smaller/larger catalytically competent minority species – their observation doesn’t necessarily affirm they are the catalyst in a system as complex as this – although this conclusion would concur with other references above as being in the appropriate size range.

Reply: As pointed out by Reviewer 1 and well known for hydrocarbon auto-oxidation, oxidation can occur without a catalyst. A free radical initiator could initiate the chain reaction. Therefore, it is essential and critical for us to establish that the solubilized Au has catalytic relevance. We believe that we have done so. Not only the catalytic relevance, the actual catalytic function of the solubilized Au is established, as detailed above. However, we are also careful in avoiding statements that we know for certain which cluster size of the solubilized Au is the most active. We described in detail that there is a distribution of cluster size. However, the activity seems to correlate with the intensity of the emission peak detected. Thus, we **established** that solubilized Au catalytically generates initiators, we only **suggested** that clusters of 7-8 atoms are likely the most active form.

Reviewers' comments:

Reviewer #1 (Remarks to the Author):

First, I would like to thank the authors for taking my points very seriously and for doing the additional experimentation. This has now led me to think about the paper in yet more detail. It is a thought provoking piece of work and the observation that clusters are being formed is potentially very important. However, while the authors feel they have supplied the key data I feel we are still not there. As I said the proposal is extraordinary and so it requires compelling evidence. I have some comments and suggestions as I am keen to see this resolved; as I am sure are the authors.

1) the cyclooctene is only 95% purity and this is worrying as there may be key impurities that play a key role. This is difficult to rule out but pretreatment of the COE may be needed. This is something that researchers often ignore. In addition does the COE contain a stabiliser?

Manufacturers do not disclose this information. However there are methods to remove this. So I suggest that the COE is cleaned up and is as pure and stabiliser free as possible.

2) hydroperoxides are only needed for Au catalysts if stabilisers are present. The hydroperoxide basically counteracts the stabiliser which is a very potent radical trap. With stabiliser free COE no peroxide initiator is required and epoxidation is observed. This has been published for Au catalysed epoxidation of cyclic and linear alkenes

3) in my initial report I asked to see the reaction product profiles with time on line. At present they show the epoxide yield and conversion. But what is needed is the allylic oxidation product profiles with time on line. My point is that an explanation is that these are formed and the alcohols are oxidised to peracids which are the effective epoxidising agent. In the filtration experiments now described these can be present and so would still be the cause. Hence I remain unconvinced that the evidence is yet compelling.

4) a suggestion would be to do a short study on cyclohexene (again making sure its stabiliser free). This is not readily epoxidised, so if the epoxide is formed then this would be compelling evidence that a new epoxidising species is being formed.

5) If the allylic product profile and the Au clusters cannot be disentangled – and this might well be the case – I am still sympathetic to publication but the conclusion would need to be softened/modified. It will still be a good paper.

Reviewer #2 (Remarks to the Author):

Referee comments on Reply to reviewers' comments, re manuscript: "Stable, Solubilized Active Au Atom Clusters for Selective Epoxidation of Cis-Cyclooctene with Molecular Oxygen" by Linping Qian, et al.

While the authors have now addressed and clarified the points around mass balance and given definitions for terms that they may regard as textbook but are commonly calculated in catalysis literature by a number of routes, not all of which result in non-cyclical mass balances. However the problems raised with comparison to the wider literature and many similar systems that needed thorough discussion have been met with a brief comment and inclusion of the references. I think given the technical similarity these are important works and need discussing to be published in a high quality journal. I also continue to hold the view that while it is an interesting report in this area, and is of likely interest to the readers of more specialist journals in the field of catalysis – as would be highlighted by proper discussion of the prior art. The authors disagree with this point but nothing now said changes my view that I do not believe the work represents a significant new step forward that justifies publication in Nature Communications.

I apologise for the confusion over the presence of solvent vs standard, but this doesn't materially alter the point being made.

Reviewer #3 (Remarks to the Author):

In this revised manuscript the authors describe the use of silica-supported gold catalysts and molecular oxygen to convert cis-cyclo-octene selectively into the epoxide. The main thrust of the work is mechanistic, however, specifically to provide evidence about the role of gold. On the basis of the evidence presented, the authors argue that, during the observed induction period, small, soluble gold clusters are formed, stabilized by oxygenated reaction products generated slowly via the supported gold nanoparticles; these soluble clusters then act as initiators of an autocatalytic cycle that produces the epoxide. The evidence I find most compelling comes from the observation that hot filtration of reaction mixtures after the induction period yields a filtrate, aliquots of which can bring about epoxidation without an induction period. The authors have carried out extensive control experiments to validate their methodology and have also quantified the role of hydroperoxides, produced in small amounts, using triphenylphosphine to quench them. Spectroscopic and EM evidence characterizing the soluble Au clusters is also presented.

I believe that the authors have produced an excellent piece of work, meticulously carried out and exhaustively reported. Their findings are of general relevance to studies of catalysis by supported metals. I recommend acceptance of the manuscript for publication.

I do have a couple of minor comments that the authors might like to consider before publication.

1. The hot filtration procedure plays a crucial role in this study and deserves to be described in more detail in the main text, perhaps with comment on its capabilities.
2. Lines 133/4: It is asserted that suberic acid or other diacids derived from 1,2-cyclo-octandiol are the probable ligands stabilizing the soluble gold clusters. Why were these readily available compounds not tested in comparison with the cyclic diol?
3. Line 307: The word "contributed" should, I think, be replaced by "attributed".

Response to Reviewers' comments

Reviewer #1

Comment 1 and Comment 2 deals with stabilizer present in the commercial cyclooctene. We address them simultaneously.

Comment 1: *The cyclooctene is only 95% purity and this is worrying as there may be key impurities that play a key role. This is difficult to rule out but pretreatment of the COE may be needed. This is something that researchers often ignore. In addition does the COE contain a stabiliser? Manufacturers do not disclose this information. However there are methods to remove this. So I suggest that the COE is cleaned up and is as pure and stabiliser free as possible.*

Comment 2. *Hydroperoxides are only needed for Au catalysts if stabilisers are present. The hydroperoxide basically counteracts the stabiliser which is a very potent radical trap. With stabiliser free COE no peroxide initiator is required and epoxidation is observed. This has been published for Au catalysed epoxidation of cyclic and linear alkenes*

Response:

We were aware of the literature information (e.g. from Hutchings' group) that stabilizers in commercial cyclooctene could affect the kinetics, and have specifically addressed this in the Supporting Information in both the original submission and the first revision, section 1.4.2. Au/SiO₂. In the initial phase of our experiments, we followed published procedure to remove the stabilizer in the cyclooctene (COE) with KOH. In the later part of the investigation, we decided to switch to purification by distillation. In our hands, generally we found only small differences between COE purified by distillation or by KOH treatment to remove the stabilizer, although purified by distillation appeared to give more consistent results. Specifically, the paragraph in SI section 1.4.2. regarding this in the original submission read:

"80 mg of 1 wt. % Au/SiO₂ was placed in a three necked flask fitted with a fine frit glass disperser tube and a condenser. The manufacturer added stabilizer (100 to 200 ppm irganox 1076) in cyclooctene was removed using 3 M KOH treatment at room temperature followed by separation and repeated washing with milli-Q water.³ using 3 M KOH treatment at room temperature followed by separation and repeated washing with milli-Q water. Sometimes this procedure was not adequate to remove all impurities, then distillation of COE was conducted using an oil bath at around 180 °C while collecting the sample at 145 °C."

We have looked for the effect of stabilizer and specifically mentioned this in the manuscript text (both original and first revision), first sentence of Results, which read:

"Solvent-free oxidation of COE using O₂ was conducted in the absence of light both with and without removal of the COE stabilizer (see Supplementary section 1.4 for details), having in mind the recent report that oxidation of commercial cyclic alkenes using O₂ over Au/graphite was only possible after removal of the added stabilizer¹⁷."

Our observations on the effect of stabilizer differed somewhat from that reported by Hutchings. As described in the original manuscript and first revision, first paragraph of Results, even with the stabilizer-free cyclooctene, there was an induction period with any of Au precursor we tested.

Using the Au/SiO₂-A catalyst, the induction period was ~2 h. Without any Au precursor, the induction period was at least 11 h. We also reported in the same paragraph that using the AuCl precursor, whether the stabilizer was removed or not did not affect the induction period. Since Hutchings' group did not investigate using different Au sources, they were unable to make these observations. In short, the situation is more complicated than what is in the literature.

Our data also showed that in the absence of stabilizer, there was an induction period **in the absence of both solubilized Au and hydroperoxide**. The induction period is eliminated by the presence of either hydroperoxide or solubilized Au. These are detailed in the first revision, reply to the reviewers' comments, and the data are shown in Figure 2 of the first revision.

In order to make it more apparent in the main text and in SI that we have purified the cyclooctene, we made the following changes in the current (second) revision:

(i) The description of the purification procedure in SI is moved to section 1.1 of SI and its section heading now reads: "Sources and Purification of Chemicals and Reagents". So that section now reads:

"1.1. Sources and Purification of Chemicals and Reagents:

AuCl (99.9%, Alfa Aesar), AuCl₃ (99.99%, Aldrich), decane ($\geq 99\%$, Sigma Aldrich), *cis*-cyclooctene (95%, Alfa Aesar) *cis*-1,2-cyclooctanediol (99%, Aldrich), cyclooctanol (99%, Aldrich), cyclooctene oxide (99%, Aldrich), 1,5-cyclooctadiene (Aldrich, $>99\%$), 1,3-cyclooctadiene (Aldrich, $>95\%$) tetrahydrofuran (Aldrich $> 99\%$), triphenylphosphine ($>98.5\%$, Sigma Aldrich), EM Quant peroxide test strips, KI (Sigma Aldrich), fumed silica (CAB-O-SIL-90, Cabot Corporation), polymethylhydrosiloxane (Sigma Aldrich) were reagents, solvents and standards used in this work.

The manufacturer added stabilizer (100 to 200 ppm irganox 1076) in cyclooctene was removed using 3 M KOH treatment at room temperature followed by separation and repeated washing with milli-Q water.¹ using 3 M KOH treatment at room temperature followed by separation and repeated washing with milli-Q water. Sometimes this procedure was not adequate to remove all impurities, then distillation of COE was conducted using an oil bath at around 180 °C while collecting the sample at 145 °C. The first fraction of the distillate was discarded."

(ii) The first sentence of Results in the main text is changed to read:

"Solvent-free oxidation of COE using O₂ was conducted in the absence of light and either with or without COE stabilizer present. Removal of the COE stabilizer was achieved either by distillation or KOH treatment (see Supplementary section 1.1 for details), having in mind the recent report that oxidation of commercial cyclic alkenes using O₂ over Au/graphite was only possible after removal of the added stabilizer¹⁷. The former method seemed to yield more consistent results."

Comment 3. *In my initial report I asked to see the reaction product profiles with time on line. At present they show the epoxide yield and conversion. But what is needed is the allylic oxidation product profiles with time on line. My point is that an explanation is that these are formed and the alcohols are oxidised to peracids which are the effective epoxidising agent. In the filtration*

experiments now described these can be present and so would still be the cause. Hence I remain unconvinced that the evidence is yet compelling.

Response:

We apologize that we missed providing the requested information. The data are now presented as Figure S-17 in Supporting Information. The figure is also reproduced below.

Small amounts of cyclooctenol and cyclooctenone were detected, and some of them could be due to decomposition of hydroperoxide in the GC.

We disagree with the reviewer that the cyclooctenol formed can be oxidized to peracid. It is much easier to oxidize cyclooctenol to cyclooctenone (i.e. secondary alcohol to ketone) than cleaving the C-C bond to form the peracid. Once ketone is formed, it cannot be easily converted to peracid. Together with good carbon balance in these experiments, we do not think that peracids are formed, at least not much.

Figure S-17. Cyclooctene conversion and yields of epoxide, cyclooctenone (c-none), cyclooctenol (c-nol), and cyclooctanediol (c-diol) as a function of reaction time, for a reaction with filtrate derived from Au/SiO₂. An amount of triphenylphosphine equivalent to 1.05 equivalent of hydroperoxide was added to the filtrate.

Comment 4. A suggestion would be to do a short study on cyclohexene (again making sure its stabiliser free). This is not readily epoxidised, so if the epoxide is formed then this would be compelling evidence that a new epoxidising species is being formed.

Response:

Another student has been studying oxidation of cyclohexene using the Au filtrate. As expected, the solubilized Au filtrate generated from the COE experiments was effective in initiating cyclohexene auto-oxidation. Also as expected, the main product formed was cyclohexene hydroperoxide, with minor amounts of epoxide, alcohol, and ketones, similar to the product distribution from auto-oxidation. Thus, it supports our model that the solubilized Au acts like an “inorganic initiator” whose role is to inject initiators to the oxidation reaction, which proceeds by auto-oxidation.

In order to further clarify this point, the last paragraph of the Result section is changed to read:

“The filtrate derived from an Au-catalyzed COE reaction was tested for cyclohexene oxidation at 60 °C using molecular O₂. After 6h, ~20% of cyclohexene was oxidized and the major product was cyclohexene hydroperoxide (~60%), with minor amounts of 2-cyclohexen-1-ol, 2-cyclohexen-1-one, and cyclohexane epoxide, a distribution similar to that from auto-oxidation^{22, 23}. Without the Au-derived filtrate, there was no reaction. Thus, the solubilized Au was capable of initiating oxidation of other cyclic alkenes.”

Comment 5: If the allylic product profile and the Au clusters cannot be disentangled – and this might well be the case – I am still sympathetic to publication but the conclusion would need to be softened/modified. It will still be a good paper.

Response:

We believe that we have disentangled the effect of allylic product and Au clusters.

Reviewer #3

Comment 1. The hot filtration procedure plays a crucial role in this study and deserves to be described in more detail in the main text, perhaps with comment on its capabilities.

Response:

In the main text, we have added the phrase “using a 0.2 µm syringe filter” when hot filtration was first mentioned on page 3, and a sentence on the limitation of the procedure immediately after that: “Although the filtration was conducted as rapidly as possible, some cooling of the hot liquid and possible precipitation in the syringe was unavoidable, although none was observed.”

Comment 2. Lines 133/4: It is asserted that suberic acid or other diacids derived from 1,2-cyclooctandiol are the probable ligands stabilizing the soluble gold clusters. Why were these readily available compounds not tested in comparison with the cyclic diol?

Response:

This is an interesting suggestion. We did not pursue the exact cause of stabilization by diol addition, since it is not the main focus of the study. We used the observation to support the proposal that the solubilized Au was stabilized by some organic ligands from by oxidation of the reactant. A definitive identification of the ligand(s) would involve another long and careful study that is in the planning stage.

Comment 3. Line 307: The word “contributed” should, I think, be replaced by “attributed”.

Response:

The correction is made. We thank the reviewer.

REVIEWERS' COMMENTS:

Reviewer #1 (Remarks to the Author):

The authors have addressed most of my concerns. While I still am not fully convinced as I said I do not want to stand in the way of publication at this time.